# Decellularized vascularized bone grafts as therapeutic solution for bone reconstruction: A mechanical evaluation

Ugo Heller[1,2,3‡]*, Robin Evrard[4,5,6‡], Benoit Lengelé[7,8], Thomas Schubert[4,5], Natacha Kadlub[1,3], Jean Boisson[2]

**1** APHP, Necker Enfant Malades, Unit of Maxillofacial Surgery and Plastic Surgery, Paris, France, **2** IMSIA, ENSTA Paris, Department of Mechanical Engineering, Palaiseau, France, **3** University Paris Cité, Paris, France, **4** Secteur des Sciences de la Santé, Institut de Recherche Expérimentale et Clinique, Neuro Musculo Skeletal Lab (NMSK), Université catholique de Louvain, Brussels, Belgium, **5** Service d'orthopédie et de Traumatologie de l'appareil locomoteur, Cliniques Universitaires Saint-Luc, Brussels, Belgium, **6** Secteur des Sciences de la Santé, Institut de Recherche Expérimentale et Clinique, Pole de Chirurgie Expérimentale et Transplantation (CHEX), Université catholique de Louvain, Brussels, Belgium, **7** Secteur des Sciences de la Santé, Institut de Recherche Expérimentale et Clinique, Pole de morphologie (MORF), Université catholique de Louvain, Brussels, Belgium, **8** Service de chirurgie plastique, Reconstructrice et esthétique, Cliniques Universitaires Saint-Luc, Brussels, Belgium

‡ UH and RE share first co-authors on this work.
* heller.ugo@gmail.com

**Data Availability Statement:** All relevant data are within the paper and its Supporting Information files.

## Abstract

### Introduction

Large bone defects are challenging for surgeons. Available reimplanted bone substitutes can't properly restore optimal function along and long term osteointegration of the bone graft. Bone substitute based on the perfusion-decellularization technique seem to be interesting in order to overcome these limitations. We present here an evaluation of the biomechanics of the bones thus obtained.

### Material and methods

Two decellularization protocols were chosen for this study. One using Sodium Dodecyl Sulfate (SDS) (D1) and one using NaOH and H2O2 (D2). The decellularization was performed on porcine forearms. We then carried out compression, three-point bending, indentation and screw pull-out tests on each sample. Once these tests were completed, we compared the results obtained between the different decellularization protocols and with samples left native.

### Results

The difference in the means was similar between the tests performed on bones decellularized with the SDS protocol and native bones for pull-out test: +1.4% (CI95% [-10.5%–12.4%]) of mean differences when comparing Native vs D1, compression -14.9% (CI95% [-42.7%– 12.5%]), 3-point bending -5.7% (CI95% [-22.5%– 11.1%]) and indentation -10.8% (CI95% [-19.5%– 4.6%]). Bones decellularized with the NaOH protocol showed different

**Funding:** This work was supported by Fondation des Gueules Cassées (Application N°41-2020, France) granted by Dr. UH The funders had no role in study design, data collection and analysis, decision to publish, or preparation of the manuscript.

results from those obtained with the SDS protocol or native bones during the pull-out screw +40.7% (CI95% [24.3%– 57%]) for Native vs D2 protocol and 3-point bending tests +39.2% (CI95% [13.7%– 64.6%]) for Native vs D2 protocol. The other tests, compression and indentation, gave similar results for all our samples.

## Conclusion

Vascularized decellularized grafts seem to be an interesting means for bone reconstruction. Our study shows that the decellularization method affects the mechanical results of our specimens. Some methods seem to limit these alterations and could be used in the future for bone decellularization.

## Introduction

Many different surgical techniques have been described for large bone defect reconstruction and several bone substitutes are currently available [1]. Thus, bone defects can be managed in different ways, depending on the general condition of the patient, as well as the localization and etiology of the bone loss. The principal etiologies of these bone defects are bone tumors, traumatic bone defects (e.g. ballistic trauma) and birth anomalies. These reconstructions are often associated with a high complication rate and in some cases multiple surgical interventions [2, 3].

Among these techniques, autologous bone grafts [4], arthroplasties, induced membrane technique with secondary bone grafting [5, 6] and vascularized bone transfers are the most commonly used for large bone defects. Those techniques can be used alone or in combination [7–9]. However, despite this wide range of reconstructive options, all these methods have drawbacks: morbidity of the donor site [10–12], infections, implant failure, fracture, non-union, necrosis and poor functional results [2, 10]. To avoid these complications, a vascularized bone allograft could allow large and complex reconstructions that could not be performed with autologous free-flaps or grafts. The developments of such vascularized allograft has been made possible over the last decades but at the cost of a heavy immunosuppressive regiment [13, 14]. Long-term immunosuppression can compromise the graft itself, but also the patient's life with the risk of serious infections, organ failure and induced cancers [15–18].

Non-vascularized bone allograft respects the anatomy of the defect and avoids donor site morbidity. Moreover, allograft could be decellularized by a combination of chemical, physical and/or enzymatic treatment [19] to avoid immunosuppressive treatment. However, preliminary studies have shown mid and long term complications such as, partial necrosis, non-union, fatigue fractures and infections [20, 21].

In this context, many research teams in tissue engineering focused their efforts on developing a bone replacement solution. Two approaches are promising: synthetic bone materials and decellularized revascularized bone grafts. Synthetic bone materials can be derived from living structures or artificially created (3D printing of matrices) and seeded with osteoinductive factors [22–28]. These methods represent an alternative to the conventional reconstruction techniques, but require an absence of immune rejection, a perfect biocompatibility, similar mechanical properties, and a good vascularization to allow a durable osteo-integration [29, 30].

Decellularized vascularized bone graft is an innovative method based on the reduction of the immunogenic content present in the tissues while retaining the tissue architecture and

non-cellular matrix. This decellularized tissue create a biological scaffold. Theoretically, this technique would allow bone reconstruction with massive bone allografts. Indeed as these tissues are cleaned of their immunogenic content, they could be recolonized by the cells of the recipient patient directly through the nutrient artery [31–36]. Biobanks of decellularized allografts could be created to perform transplants upon request, to correct large and complex bone defects thanks to vascularized bone, without donor site morbidity nor immunosuppressive therapy.

Decellularized vascularized allograft have given very promising results at the cellular level [33], but no data are currently available regarding their mechanical properties which are essential to allow a reliable and usable bone reconstruction in clinical practice. Indeed, the graft must be able to handle classic osteosynthesis and must resist to the mechanical stress of the implanted site.

The most studied decellularization protocol remains the SDS (Sodium Dodecyl Sulfate) based technique. I It had been well described at a cellular level [31–33].

A new, more aggressive protocol with NaOH (currently applied clinically for decellularization of bone allografts used primarily as bone fillers for small defects) is currently under development for vascularized massive bone allografting [37, 38]. The cellular results of this second protocol seem promising but due to the harsh physico-chemical treatment, mechanical properties of the graft could be altered.

The objective of this study was to compare the mechanical properties of bone grafts treated with these two decellularization protocols and with native bone graft using a porcine bone graft model. To our knowledge, no study of the mechanical properties of vascularly decellularized bone grafts and its pairwise comparison with native bone has been published in the literature to date.

## Materials and methods

### 1. Harvesting of bone graft

The protocol was accepted by a local ethics committee (number 2020/UCL/MD/027 and A1/UCL/2021-A1, Comité d'Ethique de Bruxelles, Belgium). All our samples were taken from 6-month-old Belgian Landrace pigs and samples were taken in the laboratory of the Institut de Recherche Expérimentale et Clinique de Louvain, Belgium. Samples from left and right forearms of each subject were harvested.

Pigs were anaesthetized with an intramuscular injection of Rompun-Zoletil (2mg/kg and 6mg/kg) and then anesthesia was maintained with isoflurane. Euthanasia was performed at the end of the procedure with an intravenous injection of T61 (1ml/5kg). The pigs were injected with a bolus of 25'000 units of intravenous heparin before euthanasia. A wide axillary skin incision allowing visualization of the pectoral muscle was made, it was continued by a frontal transpectoral incision and a dissection of the axillary fat space to reveal the vascular trunks (axillary artery and vein). Ligatures and sections of the vascular trunks proximally were performed and followed by a scapulohumeral and a radiocarpal dislocation.

Dissection was continued along the entire length of the axillary artery under magnification. Collateral vessels were ligated. Disarticulation was performed with a cold blade, staying as close to the humerus as possible, taking care to maintain the continuity of the artery.

Finally, the distal end of the artery and vein were ligated at the radiocarpal level. The brachial artery becomes the interosseous forearm artery after crossing the elbow joint. Just next to the radial head, the artery plunges between the two bones and it travels to the radiocarpal joint. Within this interosseous travel, the artery gives two main nutrient arteries: one for the ulna and one for the radius. Both nutrient arteries penetrate the bone approximately at mid length.

A patency test of the vascular pathway was performed by injecting heparinized serum to flush the vascular network and check for good venous return. The wash out was performed until the venous return came back translucent. The samples were then decellularized and stored at -80˚C.

## 2. Decellularization

This protocol was performed at the CHEX laboratory (Chirurgie Expérimentale et transplantation, IREC [Institut de Recherche expérimentale et clinique] of the UCL [Université Catholique de Louvain], Brussels, Belgium).

**Two protocols were selected in this study.** Protocol 1(D1) was performed as follow: The dissected artery was canulated and the graft was perfused with 70 L of 1% Sodium Dodecyl Sulfate (SDS, Sodium dodecyl sulfate >98%, GPR RECTAPUR, VWR, USA) to solubilize cell and nucleic membranes and denature proteins. This was followed by a phosphate saline solution (PBS) washout for 3 hours. Forty liters of 1% triton-X (Triton® X-100, grade dedicated to proteomic, VWR, USA) were then infused to disrupt DNA-protein, lipid-lipid and lipid-protein interaction for 24 hours. Grafts were rinsed with 40 L of PBS. Finally, the grafts were perfused with 25 mg of DNase (DNase I from bovine pancreas, Sigma-Aldrich®, Darmstadt, Germany), [39] dissolved in 1 L of saline at 37˚C. This last step was followed by a thorough washout with PBS for 24h Perfusion rates were controlled by a peristaltic pump at 12 mL/min with the exception of the DNase step which was performed at 4ml/min. The samples were washed with deionized water between each step.

Protocol 2 (D2) was adapted from the UC Louvain University Tissue Bank protocol. This method is based using perfusion with peristaltic pump through a canulated artery of NaOH (Sodium hydroxide solution, Sigma-Aldrich®, Darmstadt, Germany) and H2O2 (Hydrogen peroxide 30%, VWR®, Leuven, Belgium) [37, 38]. This protocol used NaOH to solubilize cytoplasmic component of the cells, and H2O2 acid to disrupt nucleic acids and denature proteins. This protocol also used Ethanol (Ethanol 96% denatured, Fagron®, Anvers, Belgium) and Acetone (Acetone, VWR®, Leuven Belgium) to lyse cells by dehydration and to solubilize and remove lipids [39]. All solutions were perfused by the means of a peristaltic pump at 12mL/min. Each perfusion step was followed by a washout with PBS. The samples were washed with deionized water between each step.

The precise description of this decellularization protocol cannot be described here, it will be the subject of a specific publication to come. This method has, however, already been used in another publication [37].

## 3. Samples preparation

To evaluate the decellularization of protocol D1, six paired pig's forearms samples were harvested. Of each pair of forearms, one was decellularized whereas the other one was kept native for comparison.

The decellularization protocol D2 was performed on 7 pig's forearms and the results obtained were studied independently. The samples decellularized with D2 had no paired native samples, but they were compared with native and D1 samples (Table 1).

Once the decellularization was completed, the olecranon was removed from the elbow for histological assessment of the decellularization (Fig 1). These analyses consisted of a DAPI (Di Aminido Phenyl lndol) a DNA coloration in fluorescent blue which allows to visualize DNA under a certain wavelength (DAPI, Sigma-Aldrich®, MO, USA), a hematoxylin and eosin coloration (HE, homemade) a coloration of cytoplasmic and of nucleic elements and a Masson's trichrome (TM, homemade) coloration of nucleic, cytoplasmic and collagen element. Those colorations were used to confirm the absence of cell and nuclei (Fig 2).

**Table 1. Sample allocations.**

|  | Pig n˚1 to 6 | Pig n˚7 to 14 |
| --- | --- | --- |
| Native left forearm | Harvest |  |
| D1 righgt forearm | Done |  |
| D2 right forearm |  | Done |
| Right forearm histological assessment | Done | Done |
| 3 Pullout tests | Performed | Performed |
| 1 Compression test | Performed | Performed |
| 1 Three point bending test | Performed | Performed |
| Indentation tests | Performed | Performed |

Table showing the decellularizations and the different tests performed on each pig. Pigs 1 to 6 had decellularization protocol n˚1 (SDS), pigs 7 to 14 had protocol n˚2 (NaOH). The mechanical tests were all performed on each of the bone samples.

The rest of the sample were frozen directly after decellularization at -80˚C. The samples were kept frozen until the mechanical tests were performed.

## 4. Mechanical tests

Screw pull-out, compression and 3 points bending tests were done with a uniaxial elongation machine (34SC-5, Instron Corp., Illinois Tool Works Inc., Glenview, IL, USA) with a 5 kN load cell (2519 series, Instron Corp., Illinois Tool Works Inc.).

Micro-indentation tests were performed using a Fischerscope HM2000® (Helmut Fischer GmbH, Sindelfingen, Germany) indenter with a Vickers tip.

For each specimen, after defrosting at room temperature for 2 hours, a radio-ulnar separation with a cold blade was done.

For each specimen we performed: 3 pullout tests on the proximal part of the radius, a compression test on the radial diaphysis, a 3-point bending test on the ulna; and multiple indentation tests after inclusion on the proximal part of the radius (Fig 3) (Table 1).

**Pull-out screw test (Fig 4).** The pull-out screw tests consist of uniaxial traction on an osteosynthesis screw screwed into the bone [40–43]. These tests were designed to assess the resistance to pullout of screws in an osteosynthesed graft.

For these tests, Stryker® osteosynthesis cortical screws (length 18 mm and diameter 2.3 mm, Stryker Inc., Kalamazoo, MI, USA, ref: 50–23418) have been used.

Holes were drilled in the samples with a 1.6 mm drill bit at a speed of 3000 rpm (Dremel 3000, Dremel Europe, Breda, Netherlands). The drilling depth was over 10 mm to ensure that

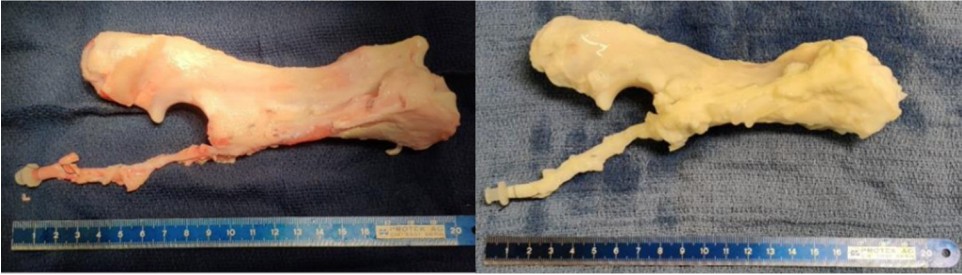

**Fig 1. Right forearm before and after decellularization.** In the picture on the left, we can see the right forearm before decellularization. The image on the right shows the appearance of the same forearm after decellularization. These pictures illustrate the white aspect of all the structures after the decellularization protocol.

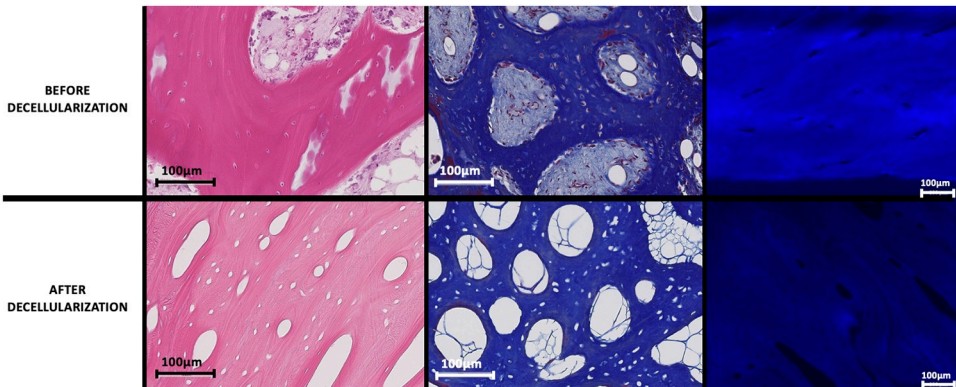

**Fig 2. Results of histologic analysis before and after decellularization.** Pictures showing the histological results before and after decellularization. The stains are from left to right: HE, TM and DAPI. The absence of DNA and nucleic elements after decellularization can be observed.

the entire cortical bone was pierced. The first six millimeters of the screw were inserted into the bone leaving 12 mm of the screw protruding. The screw head was positioned in a custom-made jaw. Once the screws were positioned, a constant speed traction test of 1 mm/min was done. The test stopped when the measured force dropped by 40% which always corresponds to the screw being pulled out.

For each specimen, this test was repeated one time on 3 separate areas (central, lateral, medial). The raw force $F$ and displacement $l$ values were extracted. The work $W$ required to pull a screw out was expressed in joules ($J$). It corresponds to the area under the Force/

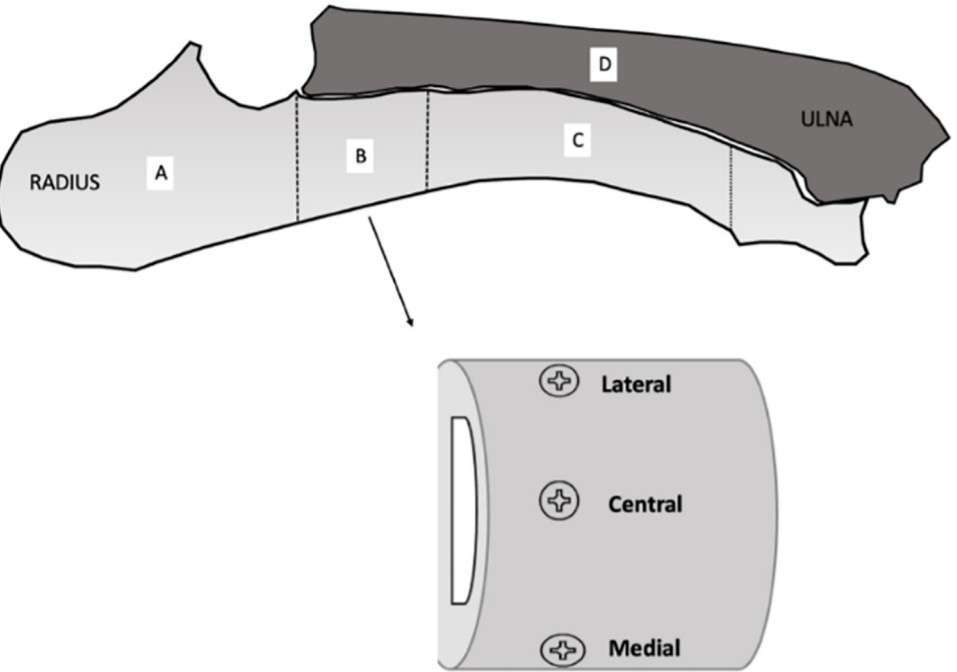

**Fig 3.** Schematic representation pig forearm; A. Part left for microscopic analysis, B. Part for Pullout and indentation, C. Part for compression, D. Ulna for 3 points-bending. This schematic representation shows the different location of the bone sample on which the tests were performed.

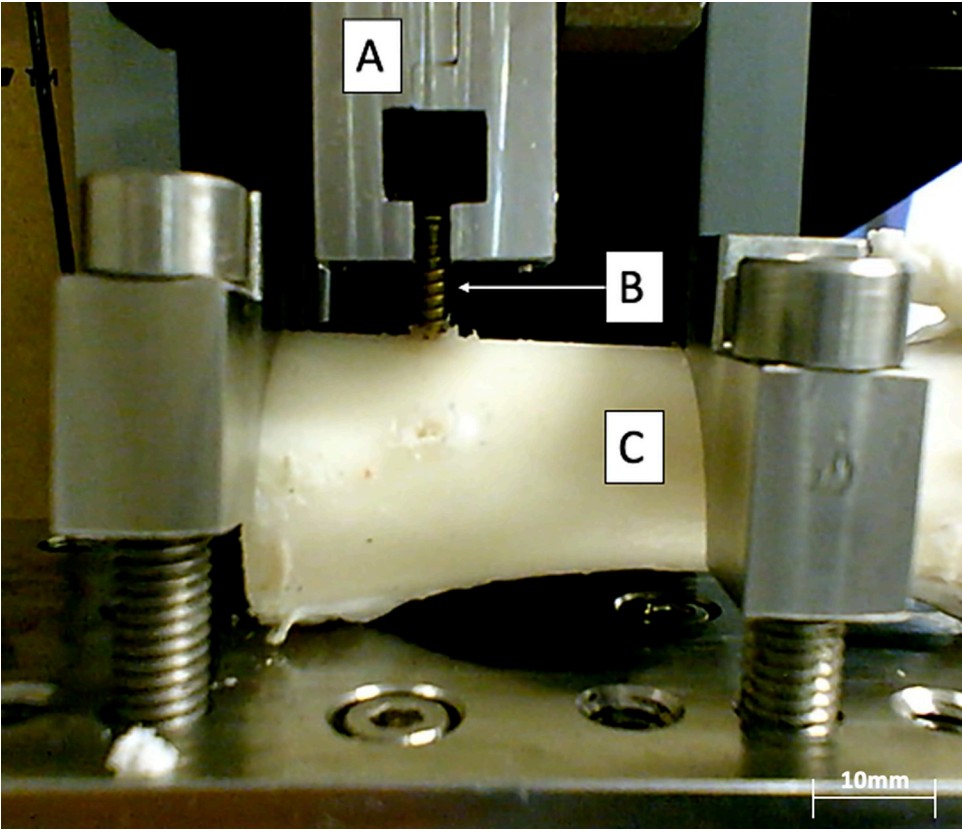

**Fig 4.** Screw pull-out tests on a decellularized bone (note the drilling of a previous screw); A. Upper Jaw, B. Screw, C. Radius. Here, the top jaw of the tensile test machine moves vertically to perform a traction force on the bone screw. The test bench fixes the position of the specimen. The bone already has a hole made from the previous tensile test on a lateral screw.

Displacement curve from the beginning of the test ($F = 0$) to the maximum force $F_{max}$ [44]:

$$W = \sum_{F=0}^{F=F_{max}} F.dl \qquad (1)$$

where $dl$ is the machine displacement increment, i.e. the machine displacement (in meter) between two force measurements.

**Compression tests (Fig 5).** For these tests, the area used for the radius screw pull-out tests was cut and the distal part of the radius was removed to obtain a bone segment similar to a radial diaphyseal cylinder (Figs 3 and 5).

The portion of the bone used for screw pullout tests was kept frozen for subsequent indentation testing.

Both sides of the bone (upper and lower) were polished using a disc polisher of 250 grit. The goal was to obtain a 30 mm long cylinder with two faces as parallel and smooth as possible for the compression test.

First, 10 cycles of compression at 0.3% deformation with a speed of 0.01 mm/s were performed. Directly after these cycles, a compression test with the same displacement rate was performed. The compression test was stopped once the local maximum was reached (decrease of more than 20% of the applied force). Test parameters were based on preliminary tests performed on non-decellularized bone. They were similar to other bone compression tests found

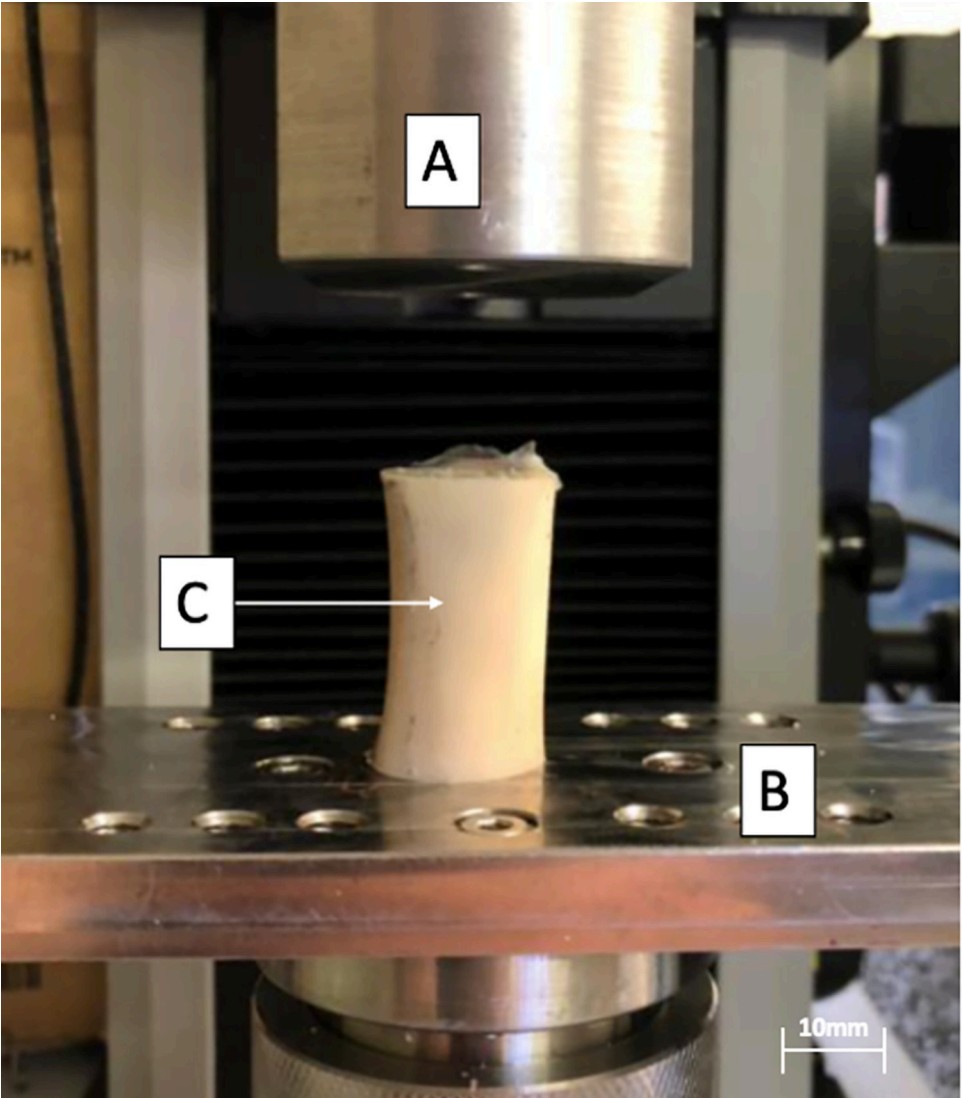

**Fig 5.** Compression test of decellularized bone (fixed bottom plate, movable top plate); A. Upper plate, B. Lower plate, C. Radial cylinder. This figure shows the installation before a compression test. The bone is placed between the two compression plates. Once the test has started, the upper plate will move down until the bone is compressed.

in literature [45–47]. Here, the raw compression force $F$ and the corresponding machine displacement $l$ values were measured.

The nominal stress was extracted, defined as a force normalized by the sample initial surface (45):

$$Stress \equiv \sigma = \frac{F}{\left(\frac{\pi R^2}{2}\right)} \tag{2}$$

where $\sigma$ is the axial stress in pascals ($Pa$), $F$ is the force in newtons ($N$) recorded by the load cell and $\left(\frac{\pi R^2}{2}\right)$ the section of a half-cylinder (in $m^2$) measured with a caliper before the test.

Strain is defined as [48]:

$$Strain \equiv \varepsilon = \frac{\Delta l}{l} \tag{3}$$

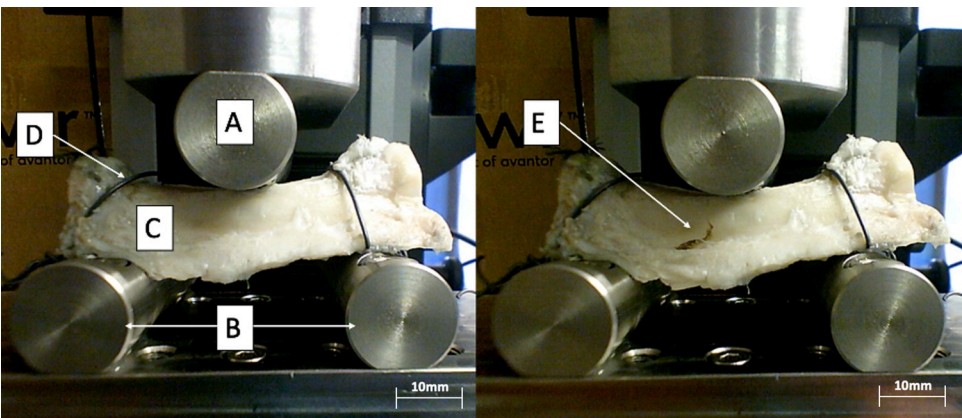

**Fig 6.** 3-point bending tests on a decellularized ulna at the beginning of the test (left) and after rupture (right); A. Upper cylinder, B. Lower cylinders, C. Ulna, D. Steel wire, E. Fracture line. The figure shows a 3-point bending test. The left picture corresponds to the test start. The upper cylinder moves down until complete fracture (right picture). The bone is fixed laterally with steel wires to prevent it from slipping.

where $\Delta l$ is the variation of the distance between the two grips measured during the tensile test and $l$ is the initial distance between the grips.

Finally, the slope of the first linear part of the stress-strain curve is extracted. This slope corresponds to the apparent elastic modulus, which approximates Young's modulus.

**3-point bending tests (Fig 6).** This test consists of applying a force on the center of bone simply supported by two cylinders. For this test, ulnas from porcine forearms were used. To prevent slippage during the test, the two ends were fixed with steel cerclage wires. The distance between the two cylinders was 50 mm for all our 3-point bending tests, making the analysis of the results more reliable. The 3-point bending tests were performed at a constant speed of 2 mm/min. These parameters were defined after performing pre-tests and using the parameters used in the literature in similar cases [46–49].

The analysis of the force-displacement curves gives:

- $F_{frac}$, the force at break (defined as the maximum force before decrease)

- $W_f$, the total work to fracture the bone, which corresponds to the area delimited by the force-displacement curve (similar to the definition of the work in section a).

**Nanoindentation (Fig 7).** The nano-indentation is a technique designed to measure the resistance of a material to localized plastic deformation [50, 51]. Here, according to standard ASTM E384-10e2, we only considered the mark left by the pyramidal undeformable tip to measure the Vickers hardness [52].

$$HV = \frac{1854.4L}{\bar{d}^2} \tag{4}$$

With $HV$, the Vickers hardness in Vickers, $\bar{d}$ is the mean diagonal length in $\mu m$, $L$ is the load in $gf$ (gram-force).

The hardness values were measured on the proximal parts of the radius in the same region as the screw pullout tests (Fig 3).

The test preparation required to cut specimens into four pieces of cortex. These pieces were embedded in resin at a temperature of 150°C and a pressure of 250 Bars, a heating time of 10 minutes followed by a cooling time of 5 minutes. The samples were then polished using three

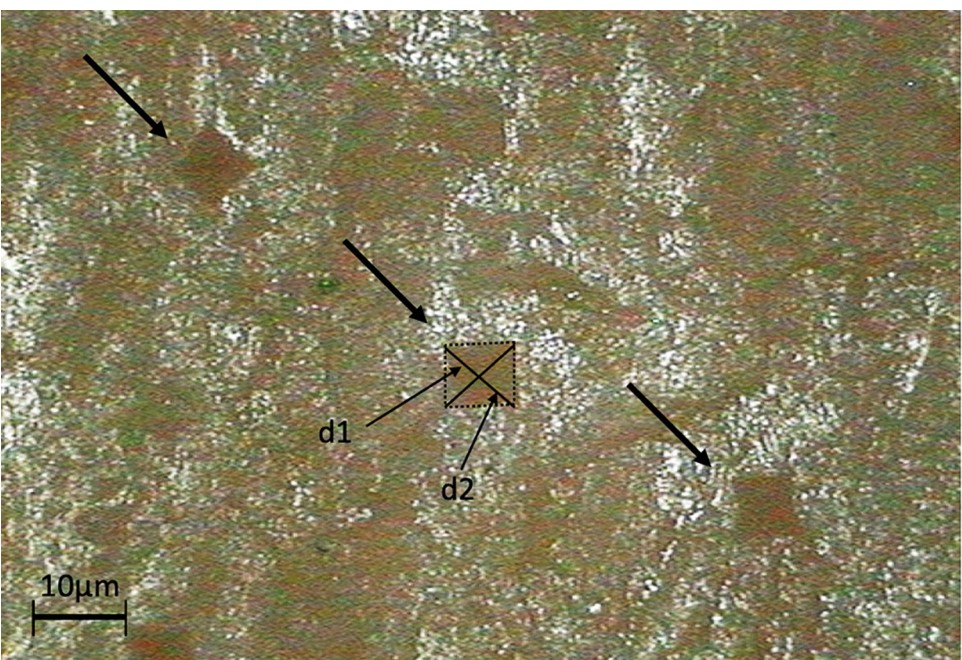

**Fig 7. Marks left by the indenter after the test (magnification x4).** Typical pyramidal marks left by the Vickers indenter. Each of these marks corresponds to a single hardness measurement.

consecutive polishing discs on an automatic polishing machine (30 seconds on an 800-grit disc, 30 seconds on a 1600-grit disc and finally 60 seconds on a 2'400-grit disc).

The tests were then performed with a force of 2N applied by the pyramidal tip on the sample with a charge and discharge time of 20 seconds separated by 5 seconds of peak time.

A minimum of 20 indentations per sample was performed to compare Vickers hardness values of native and decellularized bones.

## 5. Statistical analysis

All statistical analyses were performed on R Statistical Software (version 4.0.2; R Foundation for Statistical Computing, Vienna, Austria).

The decellularized samples (D1 and D2) were compared to the native sample. In the same manner, a comparison between each decellularized sample (D1 vs D2) was executed. The results were expressed as mean ± SD and in relative change. Relative change is the difference of means divided by the reference.

The references were natives' samples for D1 vs Native and D2 vs Native.

The references were D1 samples for D1 vs D2 comparison.

To compare the Native and D1 samples we performed a General linear mixed model (GLMM) with a random "pig" effect. This allows us to take into account in our statistical analysis that the measurements were made with matched forearms [53]. For the other analyses (D1 vs D2 and Native vs D2) we did not add a random "pig" effect, the samples samples were not matched.

Then when the measurements were repeated, we added a random "test" effect to our statistical model. This effect allows us to consider that the tests are repeated on the same samples.

Finally, if the measurements were carried out in an unrepeated manner and on unpaired samples, we performed a one-way ANOVA.

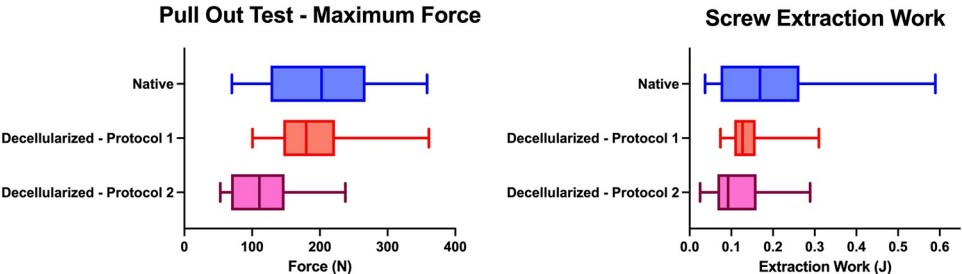

**Fig 8. Results of the pull-out test.** Left, box plots representing the extraction force (which corresponds the maximal force recorded in Newton) of the pull-out test for Native, D1 and D2. Right, box plots representing the extraction energy (in Joules) of the pull-out test for Native, D1 and D2.

No p-value will be calculated in this study because of the lack of statistical power to look for a difference or non-inferiority. The results are only reported 95% confidence interval [54, 55].

# Results

## 1. Results of the pullout tests

3 pullout tests were performed per specimen (medial, central, lateral): 18 measurements for native and D1 subjects, and 21 measurements for D2 subjects. The maximum force and the energy transmitted were extracted.

All the results are summarized in the S1 Table and Fig 8.

The means maximum forces were of 197.6 ± 79.2N in the native group, 194.8 ± 66.3N in D1, and 117.2 ± 53.0N in D2. Thus, the difference obtained was +1.4% (2.8 Newton) between the two groups during the pullout tests.

The means extraction work was 190.7 ± 143.2mJ in the native group, 145 ± 68mJ in D1, and 110.7 ± 66.7mJ in D2.

A mixed linear regression with a random "subject" (pig) effect and a random "screw" (central—lateral—medial) effect was performed to compare the results of native vs D1. A GLMM with a random "screw" effect was also performed to compare D2 with natives and with D1.

The results of this regression are presented in the following table (Table 2).

**Table 2. Result of the GLMM for the pull-out tests.**

| | Maximum Force | | | |
|---|---|---|---|---|
| Variable | Difference of means | | 95% Confidence interval | |
| | % | Newton | % | Newton |
| Native vs D1 | +1.4% | 2.8 | [-10.5%– 12.4%] | [-26.5 ; 32.2] |
| D1 vs D2 | +39.8% | 77.6 | [27.8%– 51.9%] | [54.1 ; 101.1] |
| Native vs D2 | +40.7% | 80.4 | [24.3%– 57%] | [48.1 ; 112.6] |
| | Screw extraction work | | | |
| | Difference of means | | 95% Confidence interval | |
| | % | $\times 10^{-3}$J | % | $\times 10^{-3}$J |
| Native vs D1 | +39.4% | 75.1 | [-12%– 59.8%] | [- 22.8 ; 114] |
| D1 vs D2 | +23.7% | 34.4 | [-3.9%– 51.3%] | [-5.6 ; 74.4] |
| Native vs D2 | +42% | 80 | [9.7%– 74.3%] | [18.5 ; 141.5] |

Table showing GLMM results for screw pull out tests. The results are presented as the in raw difference (in Newtons or Joules) and relative difference (in percent) means and the 95% confidence interval of the GLMM.

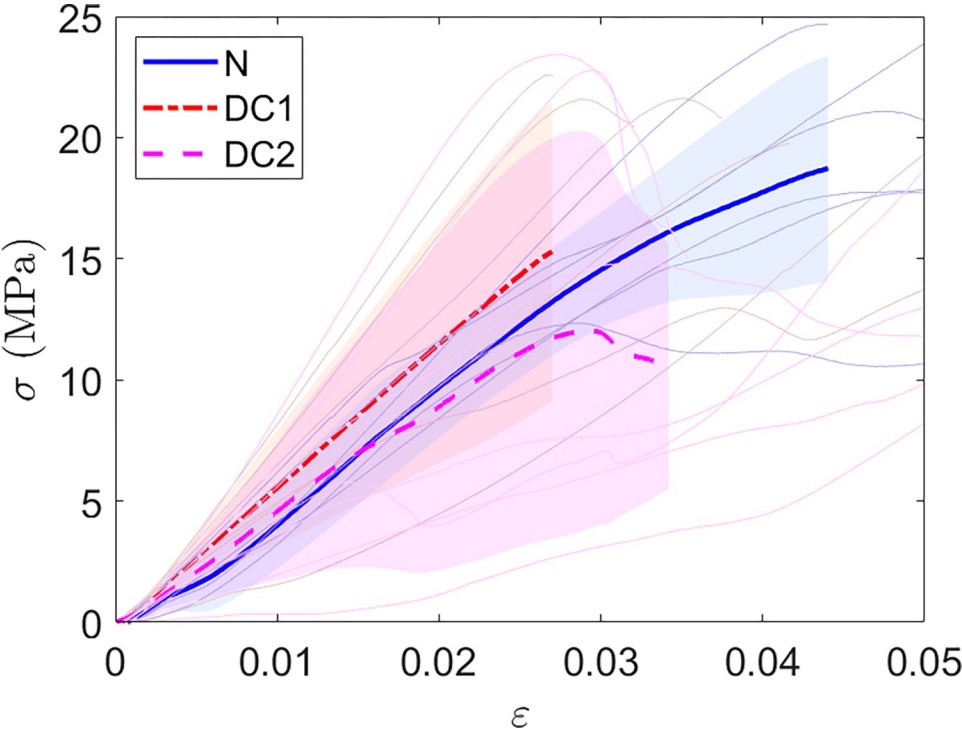

**Fig 9. Curves of compression tests.** Stress-strain curves obtained by performing compression tests. The bold curves correspond to the average curves; light areas are the standard deviation around the corresponding average curve.

The difference of means was of 1.4% (2.8N) for Native vs D1 and 40.7% (80.4N) for Native vs D2 when we focused on the maximum force. The difference of means was around 40% when we compared Native vs D1 and D2.

## 2. Results of compression tests

The compression test was carried out on 6 native and D1 cylinders and on 7 D2 cylinders.

The results of these compression tests are presented Figs 9 and 10. Interesting supplementary results can be found in in the S2 Table.

By extracting apparent elastic moduli from the different curves, a means modulus of 570.8 ± 84.8MPa for native samples, 656.9 ± 180MPa for D1, 643.4 ± 312.9MPa for D2. Our decellularized samples have a higher stiffness than our native samples.

Results of the mixed-effects linear regression with a random "subject" effect are presented in the Table 3 below with the result of the ANOVA for D2 comparisons.

Thus, a difference of -14.9% in the compression tests between the native bones and the paired D1 bones was found. This difference is comparable with the difference between the D2 and the native bones.

## 3. 3-point bending test results

Six 3-point bending tests on native and D1 decellularized ulnas were made and 7 on D2 decellularized ulnas.

The results of the tests are presented in the Figs 11 and 12 and S3 Table.

The mean maximum force was 788.1 ± 125.3N in the native group, 833.1 ± 210.4N for D1, 479 ± 190.2N for D2.

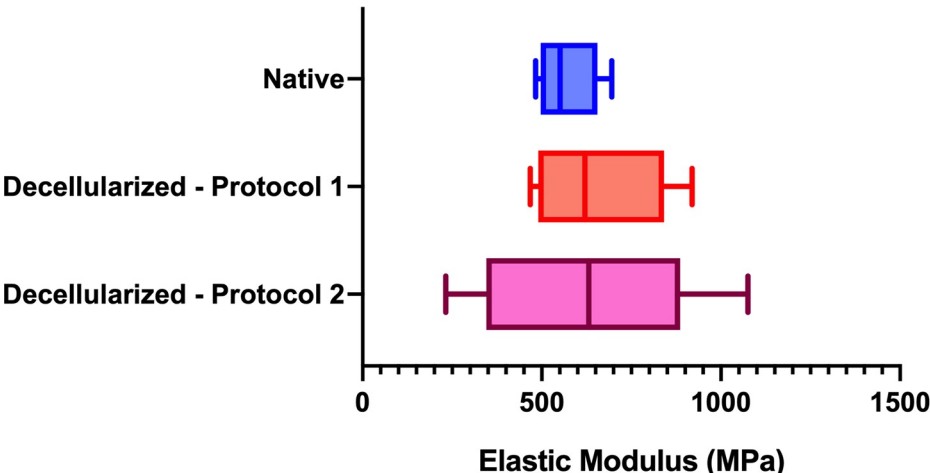

**Fig 10. Results of Compression tests.** Box plot representing the apparent elastic moduli extracted from the compression test. The apparent elastic modulus corresponds to the slope of the first linear part of the stress-strain curve.

The bending work were $2.81 \pm 0.67 \times 10^{-3}$J in the native group, $2.63 \pm 0.80 \times 10^{-3}$J in the D1 and $1.23 \pm 0.69 \times 10^{-3}$J in the D2.

A mixed linear regression with a random "subject" effect was made to compare native and D1 decellularization. For D2 versus native and versus D1 an ANOVA was performed. The results of these analyses are presented in the Table 4.

The difference was -5.7% (45 N) of maximum bending force between native and decellularized with D1 subjects. This difference is in the range of -22.5% to 11.1%.

In addition, a difference of 6% ($0.17 \times 10^{-3}$J) of bending work between native and decellularized subjects with a confidence interval of -17.8% to 30.2% was found.

The difference of means for the maximum force and for the bending work are more important when the comparison focused on D1 vs D2 and Native vs D2.

## 4. Indentation test results

Finally, 132 hardness measurements on the native samples, 152 measurements on the decellularized with D1 samples and 209 measurements for the D2 were made. We extracted the Vickers hardness measurements from each of our indentation tests for comparison.

**Table 3. Results of GLMM and ANOVA for compression tests.**

| Variable | Apparent elastic modulus (MPa) | | | |
|---|---|---|---|---|
| | Difference of means | | 95% Confidence Interval | |
| | % | MPa | % | MPa |
| Native vs D1 | - 14.9% | -85.1 | [-42.7%– 12.5%] | [-243.9 ; 71.6] |
| D1 vs D2 | +2% | 13.5 | [-46.6%– 50.7%] | [-306.2 ; 333.2] |
| Native vs D2 | -12.7% | -72.7 | [-63.8%– 38.3%] | [-364.2 ; 218.9] |

Table showing GLMM results (for Native vs D1) and ANOVA results (For D1 vs D2 and Native vs D2) for compression tests. Apparent elastic modulus is the slope of the first linear part of the stress-strain curve represented in Fig 9. The results are presented as the in raw difference (in MPa) and relative difference (in percent) means and the 95% confidence interval of the GLMM or the ANOVA.

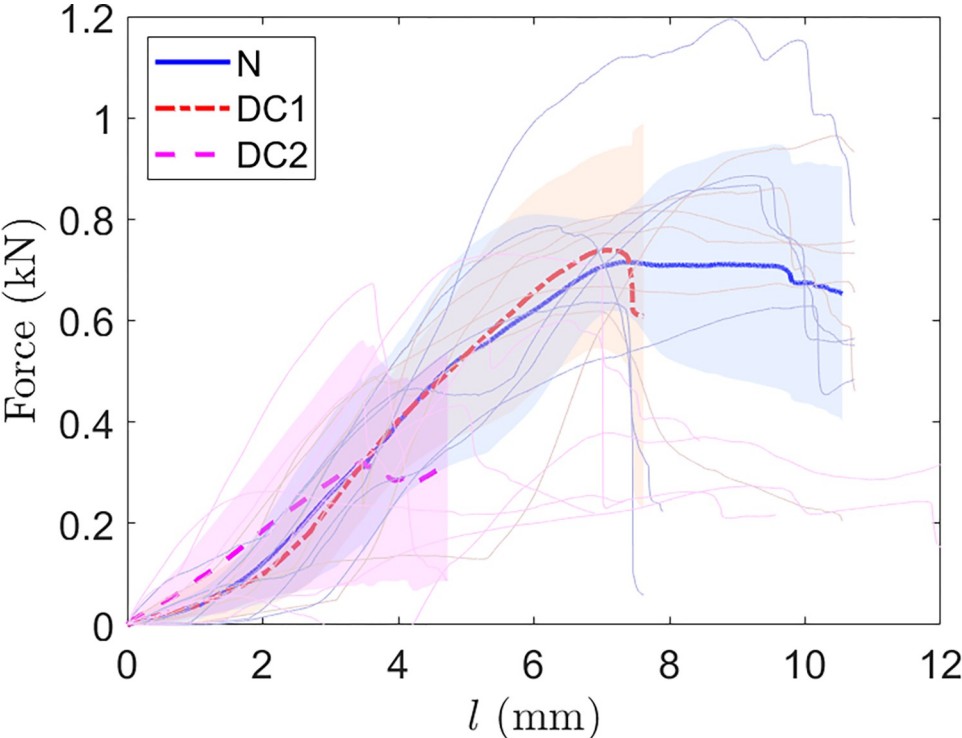

**Fig 11. Curves of 3 points bending tests.** Stress-strain curves obtained by performing 3 points bending tests. The bold curves correspond to the average curves, light areas are the standard deviation around the corresponding average curve.

The results of our indentation tests are presented in the Fig 13 and S4 Table.

The mean hardness value was 3.44 ± 2.05 HV in the native group, 3.81 ± 2.19 HV for D1 and 4.1 ± 2.91HV for D2.

The result of the linear regression with a random "subject" effect and a random "test" effect on the native vs D1 and the GLMM with random "test" effect for the D2 analysis are presented in the Table 5.

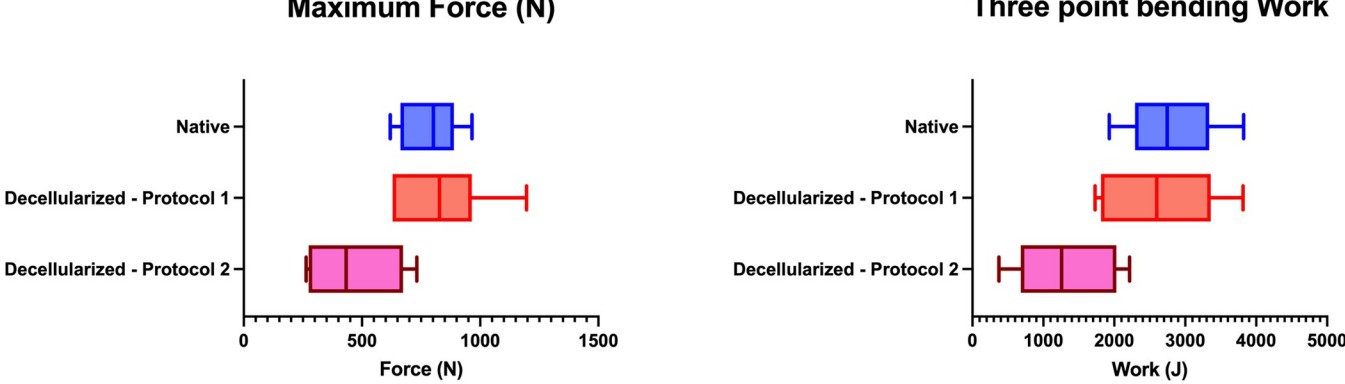

**Fig 12. Results of three-point flexion tests.** Left, box plots representing fracture force (maximal force measured in Newton) of the three-point bending test for Native, D1 and D2. Right, box plots representing the fracture energy (in Joules) of the three-point bending test for Native, D1 and D2.

**Table 4. Results of GLMM and ANOVA for 3 points bending tests.**

| | Maximum Force | | | |
|---|---|---|---|---|
| Variable | Difference of means | | 95% Confidence Interval | |
| | % | N | % | N |
| Native vs D1 | -5.7% | -45 | [-22.5%– 11.1%] | [-177.7 ; 87.8] |
| D1 vs D2 | +43.4% | 353.7 | [13.1%– 71.8%] | [109.3 ; 598.1] |
| Native vs D2 | +39.2% | 308.7 | [13.7%– 64.6%] | [108 ; 509.4] |
| | Bending Work | | | |
| | Difference of means | | 95% Confidence Interval | |
| | % | $\times 10^{-3}$J | % | $\times 10^{-3}$J |
| Native vs D1 | +6% | 0.2 | [-17.8%– 30.2%] | [-0.5 ; 0.85] |
| D1 vs D2 | +53.2% | 1.4 | [18.6%– 87.4%] | [0.5–2.3] |
| Native vs D2 | +57.1% | 1.6 | [26.4%– 85.7%] | [0.74–2.4] |

Table showing GLMM results (for Native vs D1) and ANOVA results (For D1 vs D2 and Native vs D2) for three point bending test. Results of this test are in maximum force for fracture and bending work until fracture. Results are presented as the in raw difference (in newtons or Joules) and relative difference (in percent) means and the 95% confidence interval of the GLMM or the ANOVA.

We found a difference of -10.8% with a confidence interval between -19.5% to 4.6% for native vs D1. This difference was similar to the one found when we compared D1 vs D2 and Native vs D2.

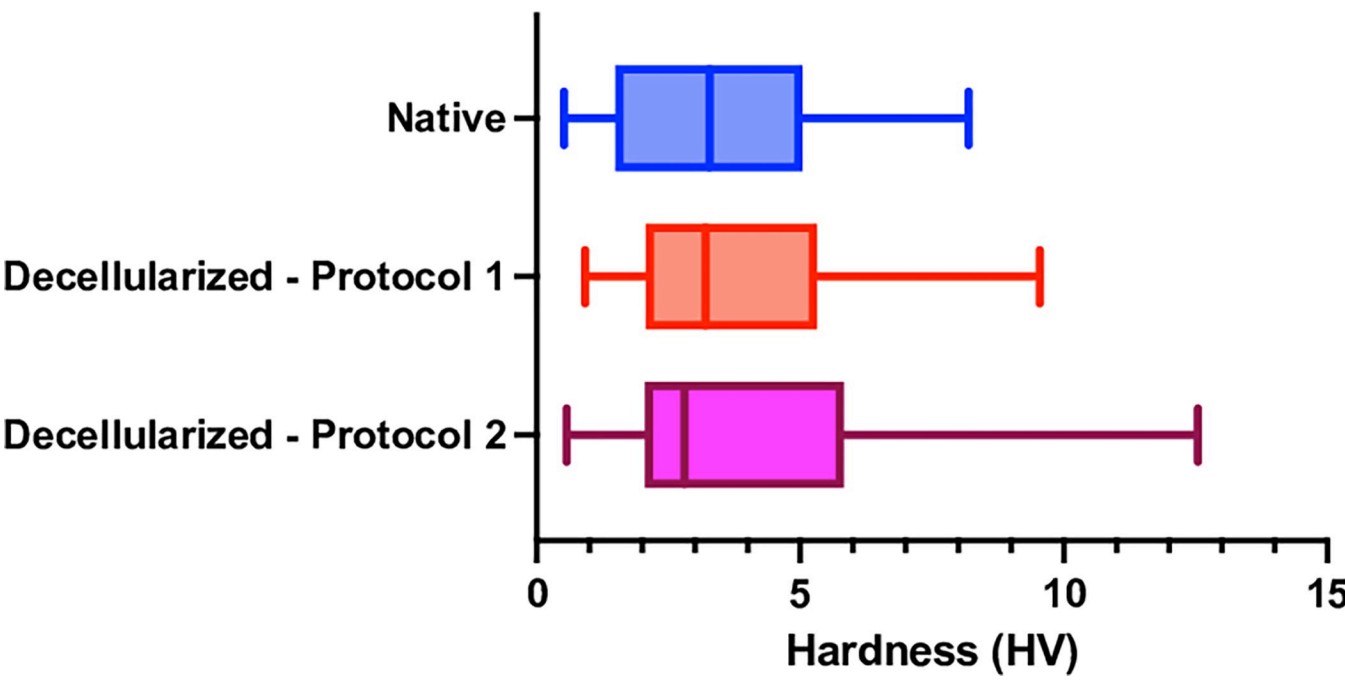

**Fig 13. Results of the indentation tests.** Box plot representing the results of the indentation test in Vickers Hardness define in Eq (4).

**Table 5. Results of GLMM for indentation tests.**

| | Hardness | | | |
|---|---|---|---|---|
| Variable | Difference of means | | 95% Confidence Interval | |
| | % | HV | % | HV |
| Native vs D1 | -10.8% | -0.37 | [-19.5%– 4.6%] | [-0.67 ; 0.16] |
| D1 vs D2 | -7.6% | -0.29 | [-22%– 7%] | [-0.84 ; 0.27] |
| Native vs D2 | -19.2% | -0.66 | [-35.8%–-2.9%] | [-1.23 ; -0.09] |

Table showing GLMM results for indentation test. Results of this test are Vickers Hardness (HV). Results are presented as the in raw difference (in HV) and relative difference (in percent) means and the 95% confidence interval of the GLMM.

## Discussion

Our study compared two decellularization by perfusion protocols in terms of bone properties with different mechanical tests (compression, pullout, 3-point bending, indentation). We started our tests with a matched comparison of native and decellularized porcine forearms using protocol D1. Then a mechanical study using a new decellularization protocol D2 (protocol NaOH) to understand if the decellularization method could affect the mechanical behavior of our bone samples.

These two decellularization protocols were chosen as they appear to be the most successful today [19, 33, 37, 38]. Numerous studies have shown the effectiveness of the SDS and NaOH decellularization protocols. Thus, our team focused on these two decellularization methods for this work [33, 34, 37, 38, 56].

SDS protocol, renamed D1 in this study, is currently used by our team to preform vascularized decellularized graft of different tissue. We choose this protocol because it is the most advanced decellularized protocol that we have. Moreover, this protocol has already shown its effectiveness in decellularizing tissues with a very significant reduction in DNA content, preservation of the extracellular matrix and promising recellularization [33], this results had been found in vitro and in vivo at a histological and cellular level.

NaOH protocol, renamed D2 in this study, has shown recent interesting results. This protocol seems to allow a more significant decrease in DNA content and thus a less inflammatory and thrombogenic recellularization [37], this results were found at histological and immunostaining level. Thus, the comparison of these two protocols seems to us to be interesting to try to choose the most adapted method of bone decellularization

For D1 versus native bone, we showed evidence of comparable mechanical properties. Confidence intervals were large but the differences in means illustrated small differences between the samples. Our observations were confirmed by the results of the GLMM. In addition, the indentation tests, the only tests that were performed with a larger sampling number, did not present differences with a confidence interval centered on 0.

We can notice that the confidence intervals are very wide for the comparison Native vs D1 and they all contain 0. This can be explained by a combination of two factors, a small number of samples associated with a small mechanical difference.

During our trials with the D2 protocol we noted some interesting differences. The 3-point bending tests and the pullout tests demonstrated a lower resistance of the decellularized bones with the D2 protocol. On the other hand, compression and indentation tests showed similar results to those obtained with D1 and native bones.

D2 thus appeared to be a more aggressive protocol at mechanical levels. It seemed to alter enough the bone matrix to make the bone more brittle compared to D1. We can thus imagine that this decellularization protocol would make the grafts more fragile for osteosynthesis and

less reliable to direct full weight bearing. It is however important to note that the D2 samples were not matched, this makes a comparative analysis with the D1 and Native samples more difficult to generalize.

Studies have already shown that fresh frozen bones versus bones defatted by detergent and then frozen seem to have no change in mechanical behavior, only irradiation seems to modify this component [57]. The irradiated bone appears more brittle, this component is caused by the loss of capacity to absorb energy in a plastic way making the bone more brittle. In addition, irradiation seems to reduce cortical bone density, which may explain this change in mechanical behavior [57].

From the mechanical aspect, the results presented here are in line with the literature for native bones. Apparent elastic modulus are close to those measured in compression tests [45–48], also the results of the pullout test were in agreement with those found in the literature [40, 42, 43]. The agreement with literature validated the bone compression and screw extraction protocol used in this article it is then possible to extrapolate the juvenile porcine bone results to a larger population. On the other hand, the results of indentation tests on native bones showed weaker resistance than those found in the literature [48, 50–52] on other kind of natives bones. This may be explained in part by the fact that the bones came from young pigs (6 months). The bone of young subjects being mineralized, this may affect the hardness of the cortical bone. The literature on bone biomechanics is very rich and the study protocols very varied, which is why in this study we tried to systematize our protocol as much as possible to make the results as reliable as possible.

The result between D1 and native tissues were also concordant with literature. Other studies have found similar biomechanical properties of grafts decellularized with a SDS protocol and native tissues. Our study of bone graft decellularized through their vascular network is in agreement with studies on other types of tissues [56, 58]. In their paper, Xu and al. studied the Young's modulus, maximum load and maximum elongation of decellularized tendon at different concentrations of SDS and Triton-X in tensile tests. They compared the results obtained with different concentrations with native tendons. Xu et al. demonstrated no statistically significant difference between decellularized tendons and their native controls [56]. Cartemell and al. also tested decellularized (with SDS) and native patellar tendons in axial tension. They compared breaking load and stiffness on stress-strain curves. They also studied the GAG content to assess whether decellularization with SDS altered the matrix. Their study showed no change between treated and native tendons in GAG content. Stiffness and peak loads were also not significantly different [58].

For NaOH and H2O2-based protocols the results are more controversial. Studies on NaOH decellularized tissues suggest that NaOH based process does not affect biomechanical properties [38]. These results differed however from other studies demonstrating mechanical difference with NaOH decellularized tissues [59]. None of the studies were conducted on bone samples. The difference that we observed for bone materials may be due to NaOH products. This product is known to cause demineralization of the cortical bone [60, 61]. In Su and al. article [60], they found that H2O2 has little to no effect on the protein content in bone. This result was also found by Uklejewski and al [61]. In another hand, they found that NaOH decreased the weight percentage of mineral and organic material in bone. Both articles suggests that the basic treatments dissolve bone mineral. Bone demineralization is an important cause of fracture as we can see in osteoporotic fractures [62, 63]. In addition, Uklejewski and al. had shown that NaOH treatment decrease bone's Young Modulus [61].

We can also suggest that D2 protocol may produce bone-cracks on the graft cortex. Su and al. showed evidence that NaOH causes cracks that are visible in electron microscopy. These changes may explain the differences in D2 results.

Extracellular matrix does not seem to be too affected by decellularization. This may explain why the indentation tests, which mainly test the matrix, did not show any differences. Furthermore, the appearance of these bone cracks does not affect the elastic part of the bone response. But we can assume that these bone cracks are fracture initiators, that causes a lower stress at break and easier large crack propagation leading to fracture. This may explain why the bone compression is not affected, as it mainly tests the elastic part. The lower maximum force and bending work during 3 point bending test and the lower maximum force and screw extraction work during pull-out test can also be explained by the presence of these cracks.

Superficial bone demineralization exposes bone growth factors trapped in the mineralized matrix. This process may act as a bone fusion catalyst. This is an interesting direction for future in vivo research. The results found on D2 could finally prove to be an advantage for bone consolidation

This study had some limitations. First, there is a large variability inherent to geometric and inter-individual differences with bio-tissues. These differences lead to large standard deviations compared to non-biological mechanical tests. Furthermore, the limited number of samples did not allow us to perform a fine analysis of the mechanical behaviors.

To overcome this, a matched comparison for our main study (Native vs. D1) was performed.

Moreover, our tests were made in a standardized way for each sample. Therefore, the results obtained, even if they are possibly biased, contains the same systematic error allowing comparison.

This matched study couldn't be done for D2 protocol, but all samples were taken from pigs of the same age, gender, race.

Further studies are needed to better understand these results. A more detailed and matched analysis should be performed between the NaOH and SDS protocols with a larger number of samples to confirm the results obtained. Recellularization is still a challenge to allow a complete graft of composite tissues like our bone grafts. Several teams are already showing promising results [64–66]. The risk of thrombosis in the vascular tree remains one of the essential complications to be dealt with to allow viable recellularization of a graft [67, 68].

In addition, the NaOH protocol will also be the subject of new in vivo mechanical tests in porcine models before and after implantation. These tests will allow to see the effect of re-cellularization on the mechanical behavior of the graft.

In conclusion, this study assessed the mechanical properties of bone grafts decellularized by two different perfusion protocols. Results showed similar biomechanical properties between native grafts and bones decellularized by a SDS based protocol. A more aggressive treatment such as the NaOH/H2O2 based protocol may affect the mechanical properties of the grafts. A choice of decellularization protocol must consider the mechanical properties of the final product and not only the efficiency at decellularizing the graft.

## Conclusion

In this work we performed several mechanical tests on decellularized and native tissues parried and unpaired. The mechanical tests were designed to represent typical mechanical load that this kind of tissue can be submitted to during and after their implantations. According to the results we obtained we can conclude that:

- • Decellularization with SDS solvent does not appear to affect the mechanical properties of the bone grafts. No statistically significant difference was found in the mechanical properties of the native bones and the bones decellularized with the SDS protocol.

- • Decellularization with NaOH and H2O2 seems to affect certain mechanical properties of the bone grafts. The average pull-out force of a screw from the bone was 80.4N lower with this protocol, the average force required to produce a fracture on the three-point bending tests was 308.7N lower than native. Compression tests and indentation tests showed no difference between the decellularized and native tissues.

In view of all these considerations we believe that vascularized decellularized grafts are a promising technique for complex reconstructive surgery. Further publications are forthcoming on the subject with in vivo recellularization tests as well as mechanical tests after recellularization.

## Supporting information

**S1 Table. Full results of the pull-out test.** Table showing all the results obtained during our pull-out tests.
(DOCX)

**S2 Table. Results of compression tests.** Table showing all the results obtained during our compression tests.
(DOCX)

**S3 Table. Results of 3 points Bending tests.** Table showing all the results obtained during our 3 points bending tests.
(DOCX)

**S4 Table. Results of the indentation tests.** Table showing all the results obtained during our indentation tests.
(DOCX)

## Author Contributions

**Conceptualization:** Ugo Heller, Benoit Lengelé, Natacha Kadlub, Jean Boisson.

**Data curation:** Ugo Heller, Robin Evrard.

**Formal analysis:** Ugo Heller.

**Investigation:** Ugo Heller.

**Methodology:** Ugo Heller, Robin Evrard, Jean Boisson.

**Supervision:** Thomas Schubert, Jean Boisson.

**Validation:** Thomas Schubert, Jean Boisson.

**Writing – original draft:** Ugo Heller, Robin Evrard.

**Writing – review & editing:** Benoit Lengelé, Thomas Schubert, Natacha Kadlub, Jean Boisson.

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
