## [Decision Letter · Decision Letter 0]

10 Aug 2022

PONE-D-22-19559Decellularized vascularized bone grafts as therapeutic solution for bone reconstruction: A mechanical evaluationPLOS ONE

Dear Dr. Heller,

Thank you for submitting your manuscript to PLOS ONE. After careful consideration, we feel that it has merit but does not fully meet PLOS ONE’s publication criteria as it currently stands. Therefore, we invite you to submit a revised version of the manuscript that addresses the points raised during the review process.

We look forward to receiving your revised manuscript.

Kind regards,

Mohammad Azadi

Academic Editor

PLOS ONE

Journal Requirements:

"The authors would like to thank the Fondation des Gueules Cassées for its help and support for this study."

"This work was supported by Fondation des Gueules Cassées (Application N°41-2020, France) granted by Dr. UH

Additional Editor Comments:

1) Quantitative results should be added to the abstract.

2) Using 42 references in the introduction makes the novelty unclear. It is better that the innovation is highlighted in the introduction, compared to the literature review.

3) All formulations need references.

4) The discussion is poor. Obtained results must be compared to other results of other similar publications.

5) The scale bar should be added to Figures 2, 4, 5, 6, and 7.

6) It is better to add a table to introduce the samples. What was the repeatability of testing? The number of specimens could be added to this new table.

Reviewers' comments:

Reviewer's Responses to Questions

**Comments to the Author**

1. Is the manuscript technically sound, and do the data support the conclusions?

Reviewer #1: Yes

Reviewer #2: Yes

2. Has the statistical analysis been performed appropriately and rigorously? 

Reviewer #1: Yes

Reviewer #2: Yes

3. Have the authors made all data underlying the findings in their manuscript fully available?

Reviewer #1: Yes

Reviewer #2: Yes

4. Is the manuscript presented in an intelligible fashion and written in standard English?

Reviewer #1: Yes

Reviewer #2: Yes

5. Review Comments to the Author

Reviewer #1: 1. Both protocols were completed by rinsing with PBS. It neutralizes the reaction of previously used detergents, however, the protocol should be completed by rinsing with deionized or distilled water to minimize the presence of detergent microelements in the graft. Maybe you used it but didn't mention it. Please add or explain if you don't use it how you confirm the complete release of detergent elements from the graft.

2. There are no legends (explanation) on figures and tables, what makes them difficult to understand.

3. Why are the figure titles in the middle of the text and not next to the figures or separated?

What does it mean on line 206 "a) Pull-out Screw Test (Fig 4)" I did not understand why it is written here. I think it is not necessary to use A and B as a test, just name it simply.

4. On line 339 (in the results) the abbreviation GLMM appears. I think you mean "General Linear Mixed Model", but it is not explained in the text. Please explain all abbreviations.

Reviewer #2: PEER PAPER REVIEW INSTRUCTIONS/TEMPLATE

Title: Biofunctionalization of 3D printed porous tantalum using Vancomycin-carboxymethyl chitosan composite coating to improve osteogenesis and anti-biofilm properties

Authors: liu, tuozhou; Xiangya

Journal / ID: am-2022-11715k

Decision: Minor Revision

Summary:

In this study, the porcine forearms were used as the experimental sites and two methods (SDS; NaOH and H2O2) were adopted to decellularize them separately. The treated samples as well as the natural porcine bone were then subjected to biomechanical tests, specifically including compression, three-point bending, indentation and screw pull-out tests.The results showed similar performance from pull-out screw, compression, 3-point bending and indentation tests carried out on bones decellularized with the SDS protocol and native bones. Bones decellularized with the NaOH protocol showed different results from those obtained with the SDS protocol or native bones during the pull-out screw and 3-point bending tests. The other tests, compression and indentation, gave similar results for all their samples.This article shows that decellularization can have an effect on the biomechanical properties of natural bone and provides insights into which methods might reduce this effect.

Critique:

General Comments

• Materials and Methods:

After bone grafting at the site of bone defect, the biomechanical properties of the implant may be altered due to the in vivo biological environment. And this of the biomechanical performance of the decellularized sample in the organism precisely determines whether the defect site has a good osseointegration effect or not. Therefore, it is suggested that the authors may consider the inclusion of in vivo experiments.

•Discussion

In this article, Sodium Dodecyl Sulfate (SDS) and NaOH and H2O2 substitute based on the perfusion-decellularisation technique were chosen as the two variables of this experiment, so the interactions between these two variables and more relevant research are suggested to add into the discussion section, in order to make this manuscript more logical and complete.

Specific Comments:

•Abstract

Line44: A full stop is missing in the last sentence, please add it.

• Materials and Methods:

Line 160-166:The operation steps of Protocol 2(D2) are not written in detail.

Line 220:It is not stated exactly how many times the experiment was repeated.

• Results：

Line 387：Is there supposed to be a minus sign in front of 72.7.

6. PLOS authors have the option to publish the peer review history of their article (what does this mean?). If published, this will include your full peer review and any attached files.

Reviewer #1: No

Reviewer #2: No

---

## [Author Response · Author response to Decision Letter 0]

14 Sep 2022

Dear Editor, Dear Reviewers, 

We thank you for having considered our paper entitled: Decellularized vascularized bone grafts as therapeutic solution for bone reconstruction: A mechanical evaluation.

We would like to sincerely thank you for your relevant comments and questions that will significantly help us to enhance the quality of the article. 

In the following section, we respond to specific comments from reviewers. We also pointed in the article the main changes, that we colored according to one who induced it: Academic editor, Reviewer #1, Reviewer #2.

We thank you for considering another time this reviewed manuscript.

Academic editor,

We tried to ensure that our manuscript was as close as possible to the recommendations made by PLOS ONE. The article should now meet PLOS ONE's style requirements.

Thank you for pointing out this lack. We have added this to our manuscript

"The authors would like to thank the Fondation des Gueules Cassées for its help and support for this study." We note that you have provided funding information that is not currently declared in your Funding Statement. However, funding information should not appear in the Acknowledgments section or other areas of your manuscript. We will only publish funding information present in the Funding Statement section of the online submission form. 

"This work was supported by Fondation des Gueules Cassées (Application N°41-2020, France) granted by Dr. UH

We apologize for this confusion. We have removed this part in Acknowledgments. The Funding Statement complete, now, your recommendations

This was modified and added in the manuscript.

5. Quantitative results should be added to the abstract.

We added quantitative results in the manuscript.

6. Using 42 references in the introduction makes the novelty unclear. It is better that the innovation is highlighted in the introduction, compared to the literature review.

We have reduced the number of references in the introduction and try to express the novelty of our work by pointing out that this is the first time that this kind of mechanical comparison is performed. We also have tried to highlight the novelty of our paper in the introduction.

7. All formulations need references.

Added in the manuscript.

8. The discussion is poor. Obtained results must be compared to other results of other similar publications.

Thank you for your comment. We have added comparisons with the literature to enrich our discussion, but as it is the first time this kind of comparison is done the literature is poor on the subject.

9. The scale bar should be added to Figures 2, 4, 5, 6, and 7.

Done.

10. It is better to add a table to introduce the samples. What was the repeatability of testing? The number of specimens could be added to this new table.

Thank you for this relevant comment. We added a table to make sample allocations and repeatability easier for the reader to understand. 

Reviewer #1: 

1. Both protocols were completed by rinsing with PBS. It neutralizes the reaction of previously used detergents; however, the protocol should be completed by rinsing with deionized or distilled water to minimize the presence of detergent microelements in the graft. Maybe you used it but didn't mention it. Please add or explain if you don't use it how you confirm the complete release of detergent elements from the graft.

Thank you for bringing this to our attention. Indeed, we did wash with deionized water between each stage. We have added this point to the text (line 163 and line 170).

2. There are no legends (explanation) on figures and tables, what makes them difficult to understand.

We have added legends to the figures and tables to make them more understandable to the reader.

3. Why are the figure titles in the middle of the text and not next to the figures or separated?

We have tried to follow the publisher's recommendations. The figures are on a separate file and the titles and legends are in the manuscript at the request of the editor. We have uploaded a version of the manuscript with images within it for better readability.

4. What does it mean on line 206 "a) Pull-out Screw Test (Fig 4)" I did not understand why it is written here. I think it is not necessary to use A and B as a test, just name it simply.

Done.

5. On line 339 (in the results) the abbreviation GLMM appears. I think you mean "General Linear Mixed Model", but it is not explained in the text. Please explain all abbreviations.

Thank you for pointing that out. We corrected it in the text.

Reviewer #2: 

1. After bone grafting at the site of bone defect, the biomechanical properties of the implant may be altered due to the in vivo biological environment. And this of the biomechanical performance of the decellularized sample in the organism precisely determines whether the defect site has a good osseointegration effect or not. Therefore, it is suggested that the authors may consider the inclusion of in vivo experiments.

Dear reviewer, thank you for this pertinent comment. We are currently planning an in vivo study on bone grafts decellularized by vascular injection. This study will be conducted over the coming year and will be the subject of an article on the histological, mechanical, and biological results.

2. In this article, Sodium Dodecyl Sulfate (SDS) and NaOH and H2O2 substitute based on the perfusion-decellularisation technique were chosen as the two variables of this experiment, so the interactions between these two variables and more relevant research are suggested to add into the discussion section, in order to make this manuscript more logical and complete.

The SDS and NaOH protocols are currently the most promising in our research area. Most of the recent articles on decellularization focus on these methods of decellularization by varying the concentrations of these products. We have added a section in the discussion to try to explain our choice to the readers.

3. Line44: A full stop is missing in the last sentence, please add it.

Done.

4. Line 160-166: The operation steps of Protocol 2(D2) are not written in detail.

The detailed protocol is the subject of a specific article currently in redaction.

5. Line 220: It is not stated exactly how many times the experiment was repeated.

This was missing from our article and was also pointed out by the editor. We have therefore added a paragraph to our article and a descriptive table. We hope to have made the information more understandable.

6. Line 387：Is there supposed to be a minus sign in front of 72.7.

Thank you for the remark. We have corrected in the manuscript

---

## [Decision Letter · Decision Letter 1]

3 Oct 2022

PONE-D-22-19559R1Decellularized vascularized bone grafts as therapeutic solution for bone reconstruction: A mechanical evaluationPLOS ONE

Dear Dr. Heller,

Thank you for submitting your manuscript to PLOS ONE. After careful consideration, we feel that it has merit but does not fully meet PLOS ONE’s publication criteria as it currently stands. Therefore, we invite you to submit a revised version of the manuscript that addresses the points raised during the review process.

We look forward to receiving your revised manuscript.

Kind regards,

Mohammad Azadi

Academic Editor

PLOS ONE

Journal Requirements:

Additional Editor Comments:

1) Figure 2 still has no scale bar. It must be mention on the image and not in the title.

2) All formulations need references, unless they were extracted by the authors.

3) The discussion is still poor. The authors do not need to find the exact topic! The results could be trendly compared to other similar results. These results are in the references that the authors have used them for a comparison in the literature.

4) Using some sentences about that there is no literature for the discussion is not proper and they must be removed or rewritten in a proper state.

Reviewers' comments:

Reviewer's Responses to Questions

**Comments to the Author**

1. If the authors have adequately addressed your comments raised in a previous round of review and you feel that this manuscript is now acceptable for publication, you may indicate that here to bypass the “Comments to the Author” section, enter your conflict of interest statement in the “Confidential to Editor” section, and submit your "Accept" recommendation.

Reviewer #2: (No Response)

2. Is the manuscript technically sound, and do the data support the conclusions?

Reviewer #2: Yes

3. Has the statistical analysis been performed appropriately and rigorously? 

Reviewer #2: Yes

4. Have the authors made all data underlying the findings in their manuscript fully available?

Reviewer #2: Yes

5. Is the manuscript presented in an intelligible fashion and written in standard English?

Reviewer #2: Yes

6. Review Comments to the Author

Reviewer #2: Based on perfusion-decellularized technique, this article compared the biomechanical differences between decellularized vascularized bone grafts prepared by different decellularization protocols and untreated samples. Overall, there are some flaws existing. I hope these suggestions will be helpful, and the article will meet the publication standards after revision.

1. This article proves that the decellularization method affects the mechanical results. However, the description of the decellularization method is vague. A detailed description is needed, such as the duration and frequency of each step, the end point of decellularization. These may also have some influence on the biomechanical properties. It is better to supplement this information.

2. Line 193: Fig 2. The results of histologic analysis before decellularization are blurred and unclear. In addition, it would be more convincing if the results of histological obtained by different decellularization methods were compared with those before decellularization.

3. Line 214: The annotations of Table 1 does not indicate the meaning of X properly.

4. The discussion of biomechanical differences between the two methods is not sufficient. In addition, Line 544-546, "NaOH and H2O2 cause demineralization of the cortical bone (57)". The presentation does not correspond to the original text of the literature.

7. PLOS authors have the option to publish the peer review history of their article (what does this mean?). If published, this will include your full peer review and any attached files.

Reviewer #2: No

---

## [Author Response · Author response to Decision Letter 1]

15 Nov 2022

Dear Editor, Dear Reviewers,

We thank you for having considered our paper entitled: Decellularized vascularized bone grafts as therapeutic solution for bone reconstruction: A mechanical evaluation.

We would like to thank you for your comments and questions that will help us to enhance the quality of our article.

In the following section, we respond to specific comments from reviewers. We also pointed in the article the main changes, that we colored according to one who induced it: Academic editor, Reviewer #1.

We thank you for considering another time this reviewed manuscript.

Academic editor,

1. Figure 2 still has no scale bar. It must be mention on the image and not in the title.

We have changed figure 2 to a more readable figure with a scale bar included

2. All formulations need references unless they were extracted by the authors.

Thank you for pointing out this lack. We have added this to our manuscript.

However, we are not sure what you mean by "formulation".

We understood that

(1) we had to put the references of the products used: ex DNase I from bovine pancreas, Sigma- AldrichÒ, Darmstadt, Germany

(2) we have also made a call in the text to the mathematical formulas (in the previous revision) ex L245

3. The discussion is still poor. The authors do not need to find the exact topic! The results could be trendly compared to other similar results. These results are in the references that the authors have used them for a comparison in the literature.

We have tried to support our discussion by adding comparisons with the literature on the subject.

4. Using some sentences about that there is no literature for the discussion is not proper and they must be removed or rewritten in a proper state.

We changed this part when we modified the discussion.

We have checked all our references and have not found any retracted items. However, we have added a reference from Ulklejewski (n°58 in our article) which we found interesting to illustrate our discussion.

Reviewer #1:

1. This article proves that the decellularization method affects the mechanical results. However, the description of the decellularization method is vague. A detailed description is needed, such as the duration and frequency of each step, the end point of decellularization. These may also have some influence on the biomechanical properties. It is better to supplement this information.

Thank you for the notification. Decellularization via the H2O2 process is not fully described in our article. We currently do not have the rights to publish this method, which will be published shortly in a full article on the subject. This decellularization method has already been published in 2018 (Van Steenberghe and al, Annals of Vascular Surgery, 2018).

We know that this is an important gap in our paper as the concentrations and infusion times can change our results.

2. Line 193: Fig 2. The results of histologic analysis before decellularization are blurred and unclear. In addition, it would be more convincing if the results of histological obtained by different decellularization methods were compared with those before decellularization.

We have modified this figure with a more readable and clearer one. We have not added an image according to the type of decellularization (SDS vs H2O2) because the differences are too small in photographic level.

3. Line 214: The annotations of Table 1 does not indicate the meaning of X properly.

We modified this part in the manuscript.

4. The discussion of biomechanical differences between the two methods is not sufficient. In addition, Line 544-546, "NaOH and H2O2 cause demineralization of the cortical bone (57)". The presentation does not correspond to the original text of the literature.

Thank you for notifying us of this. We have modified the part concerning this reference and we have expanded the conclusion of this article to make it more precise.

---

## [Editor Report · Decision Letter 2]

28 Nov 2022

PONE-D-22-19559R2Decellularized vascularized bone grafts as therapeutic solution for bone reconstruction: A mechanical evaluationPLOS ONE

Dear Dr. Heller,

Thank you for submitting your manuscript to PLOS ONE. After careful consideration, we feel that it has merit but does not fully meet PLOS ONE’s publication criteria as it currently stands. Therefore, we invite you to submit a revised version of the manuscript that addresses the points raised during the review process.

We look forward to receiving your revised manuscript.

Kind regards,

Mohammad Azadi

Academic Editor

PLOS ONE

Journal Requirements:

Additional Editor Comments:

Before the acceptance of the revised article, a further revision should be done based on the following comments.

1) Formulations mean equations! The authors have 4 formulation for stress, strain, etc. All they need references.

2) The authors mentioned "We currently do not have the rights to publish this method, which will be published shortly in a full article on the subject. This decellularization method has already been published in 2018 (Van Steenberghe and al, Annals of Vascular Surgery, 2018).". All these description should be fully described in the text. Moreover, the future work could be mentioned at the end part of the conclusion.

3) No references could be found for 2022 and 2021! This issue is not acceptable for a scientific article, which references are not updated.

4) No separated part could be found for the conclusion! Moreover, this part should be rewritten one by one, in bullets, to show the novelty with quantitative data.

5) The standard should be mentioned for all testing types.

6) No quantitative results could be found in the abstract.

7) The quality of Figure 7 is not proper.

---

## [Author Response · Author response to Decision Letter 2]

21 Dec 2022

Dear Editor, 

We thank you for having considered our paper entitled: Decellularized vascularized bone grafts as therapeutic solution for bone reconstruction: A mechanical evaluation.

We would like to thank you for your comments and questions that will help us to enhance the quality of our article. 

In the following section, we respond to specific comments from the Academic Editor. We also pointed in the article the main changes, that we colored in blue. 

We thank you for considering another time this reviewed manuscript.

1. Formulations mean equations! The authors have 4 formulations for stress, strain, etc. All they need references

Sorry for this confusion. We added references to all the equations cited in our paper.

2. The authors mentioned "We currently do not have the rights to publish this method, which will be published shortly in a full article on the subject. This decellularization method has already been published in 2018 (Van Steenberghe and al, Annals of Vascular Surgery, 2018).". All these description should be fully described in the text. Moreover, the future work could be mentioned at the end part of the conclusion

Thank you for this comment. We tried to describe the protocol with as much details as possible in the manuscript and we also added a sentence in the conclusion on the future work. 

3. No references could be found for 2022 and 2021! This issue is not acceptable for a scientific article, which references are not updated.

We have updated our references to include the latest articles on the subject. 

For this reason, we have added the following articles: 

- Parmaksiz M, Elçin AE, Elçin YM. Biomimetic 3D-Bone Tissue Model. Methods Mol Biol. 2021;2273:239‑50.

- Wüthrich T, Lese I, Haberthür D, Zubler C, Hlushchuk R, Hewer E, et al. Development of Vascularized Nerve Scaffold Using Perfusion-Decellularization and Recellularization ». Materials Science & Engineering C, Materials for Biological Applications. déc 2020;117:111311. 

- Gerli MFM, Guyette JP, Evangelista-Leite D, Ghoshhajra BB, Ott HC. Perfusion decellularization of a human limb: A novel platform for composite tissue engineering and reconstructive surgery. PLoS One. 2018;13(1):e0191497.

- Kurokawa S, Hashimoto Y, Funamoto S, Murata K, Yamashita A, Yamazaki K, et al. In vivo recellularization of xenogeneic vascular grafts decellularized with high hydrostatic pressure method in a porcine carotid arterial interpose model. PLOS ONE. 22 juill 2021;16(7):e0254160. 

- Fooladi S, Faramarz S, Dabiri S, Kajbafzadeh A, Nematollahi MH, Mehrabani M. An efficient strategy to recellularization of a rat aorta scaffold: an optimized decellularization, detergent removal, and Apelin-13 immobilization. Biomater Res. 22 sept 2022;26(1):46.

- Seiffert N, Tang P, Keshi E, Reutzel-Selke A, Moosburner S, Everwien H, et al. In vitro recellularization of decellularized bovine carotid arteries using human endothelial colony forming cells. J Biol Eng. 21 avr 2021;15(1):15.

- Wang X, Chan V, Corridon PR. Decellularized blood vessel development: Current state-of-the-art and future directions. Front Bioeng Biotechnol. 2022;10:951644. 

- Adil A, Xu M, Haykal S. Recellularization of Bioengineered Scaffolds for Vascular Composite Allotransplantation. Front Surg. 25 mai 2022;9:843677. 

4. No separated part could be found for the conclusion! Moreover, this part should be rewritten one by one, in bullets, to show the novelty with quantitative data.

We have added a conclusion with bullet points to our article. 

5. The standard should be mentioned for all testing types.

Apart from the indentation tests - to which we added the standard reference - the mechanical tests we performed are not part of any standard procedure because they were applied to non-standardized samples. Therefore, no standardization process could be applied, even if the lab is ISO9001 certified.

6. No quantitative results could be found in the abstract.

We have changed our abstract to include more quantitative results

7. The quality of Figure 7 is not proper.

We have used a new figure. We wish this one to be more suitable.

---

## [Editor Report · Decision Letter 3]

22 Dec 2022

Decellularized vascularized bone grafts as therapeutic solution for bone reconstruction: A mechanical evaluation

PONE-D-22-19559R3

Dear Dr. Heller,

We’re pleased to inform you that your manuscript has been judged scientifically suitable for publication and will be formally accepted for publication once it meets all outstanding technical requirements.

Kind regards,

Mohammad Azadi

Academic Editor

PLOS ONE

Additional Editor Comments (optional):

It is done.
---

## [Editor Report · Acceptance letter]

5 Jan 2023

PONE-D-22-19559R3 

Decellularized vascularized bone grafts as therapeutic solution for bone reconstruction: A mechanical evaluation 

Dear Dr. Heller:

I'm pleased to inform you that your manuscript has been deemed suitable for publication in PLOS ONE. Congratulations! Your manuscript is now with our production department. 

Kind regards, 

on behalf of

Dr. Mohammad Azadi 

Academic Editor

PLOS ONE